# Virus-Induced Gene Silencing in the Tea Plant (*Camellia sinensis*)

**DOI:** 10.3390/plants12173162

**Published:** 2023-09-03

**Authors:** Wei Yang, Xianya Chen, Jiahao Chen, Peng Zheng, Shaoqun Liu, Xindong Tan, Binmei Sun

**Affiliations:** College of Horticulture, South China Agricultural University, Guangzhou 510642, China; 13202849106@163.com (W.Y.); xianya0211@163.com (X.C.); cjhtea@stu.scau.edu.cn (J.C.); zhengp@scau.edu.cn (P.Z.); scauok@scau.edu.cn (S.L.)

**Keywords:** virus-induced gene silencing, *Camellia sinensis*, tobacco rattle virus (TRV), phytoene desaturase (PDS)

## Abstract

The recent availability of a number of tea plant genomes has sparked substantial interest in using reverse genetics to explore gene function in tea (*Camellia sinensis*). However, a hurdle to this is the absence of an efficient transformation system, and virus-induced gene silencing (VIGS), a transient transformation system, could be an optimal choice for validating gene function in the tea plant. In this study, phytoene desaturase (PDS), a carotenoid biosynthesis gene, was used as a reporter to evaluate the VIGS system. The injection sites of the leaves (leaf back, petiole, and stem) for infiltration were tested, and the results showed that petiole injection had the most effective injection, without leading to necrotic lesions that cause the leaves to drop. Tea leaves were inoculated with *Agrobacterium* harboring a tobacco rattle virus plasmid (pTRV2) containing a *CsPDS* silencing fragment. The tea leaves exhibited chlorosis symptoms 7–14 days after inoculation, depending on the cultivar. In the chlorosis plants, the coat protein (CP) of tobacco rattle virus (TRV) was detected and coincided with the lower transcription of *CsPDS* and reduced chlorophyll content compared with the empty vector control, with 81.82% and 54.55% silencing efficiency of ‘LTDC’ and ‘YSX’, respectively. These results indicate that the VIGS system with petiole injection could quickly and effectively silence a gene in tea plants.

## 1. Introduction

Tea (*Camellia sinensis* L.) is one of the most important non-alcoholic beverages worldwide, and it continues to gain popularity due in part to its health benefits. Tender tea leaves contain more than 700 chemicals, among which flavonoids, amino acids, vitamins (C, E, K), caffeine, and polysaccharides are linked to improved health [1,2]. It has been reported that tea lowers cholesterol and prevents cardiovascular disease, influenza virus infection, human cancer, and so on [3,4,5]. In recent years, many genomic and transcriptomic data of *C. sinensis* have become available, but the functional characterization of genes lags behind that of other important cash crops [6,7,8,9]. One of the primary reasons is the lack of a highly efficient transformation system, which greatly restricts research on gene function in tea plants. Like most woody plants, the regeneration of *C. sinensis* is extremely difficult to achieve [1]. Therefore, developing a high-throughput and efficient in vivo method for gene function analysis in tea plants is critical. Virus-induced gene silencing (VIGS) is a simple and straightforward transient transformation method for reverse genetics, i.e., suppressing a gene’s expression to characterize its function [10]. VIGS is a natural plant defense mechanism used to combat viral infection. When the viral vector enters the host plant for the first time, an immune response is elicited by the host plant, wherein it produces an RNA-induced silencing complex (RISC). Homologous viral RNA specifically interacts with RISC in the cytoplasm, leading to its destruction. Transcriptional gene silencing or post-transcriptional gene silencing is the end result [11,12,13]. VIGS offers an attractive and quick alternative for knocking out the expression of a gene without the need to genetically transform the plant. Therefore, it has been widely applied to study gene function in crops that are difficult to establish stable genetic transformation systems for, such as pepper (*Capsicum annuum* L.), pear (*Pyrus* spp.), peach (*Prunus persica* L.), litchi (*Litchi chinensis*), soybean (*Glycine max* L.), cotton (*Gossypium* spp.), and apple (*Malus pumila*) [14,15,16,17,18,19].

The *PDS* gene is a very stable gene and widely exists in a variety of plants encoding the synthetase of phytoene desaturase (PDS). PDS catalyzes the production of carotenoids; when it is silenced in plants, chlorophyll is no longer under the protection of carotenoids, resulting in photobleaching and chlorosis of the leaves. Due to this straightforward and obvious phenotype, *PDS* is widely used as an indicator of the success of VIGS [10,20,21]. Moreover, this obvious phenotype could well predict the time of the silencing effect of the target gene and the duration of the silencing system in the recipient plant. Kumaga et al. [22] first applied VIGS technology to plants by inserting a tobacco *PDS* fragment into tobacco mosaic virus (TMV), which resulted in the epistatic degradation of chlorophyll. Its leaves were chlorotic and whitened, accompanied by a significant downregulation of *PDS*. However, the duration of TMV-induced silencing was short, which limited its application. Viral vectors are indispensable to the VIGS system and are the vital media for achieving gene silencing in the VIGS system. Through modifying the virus according to its own characteristics, the success rate of VIGS system experiments and the silencing efficiency of the VIGS system greatly improve. Tobacco rattle virus (TRV) is a soil-borne RNA virus with a straight, rod-shaped genome consisting of two parts: RNA1 and RNA2. RNA1 contains the genes necessary for viral replication and transport, and RNA2 encodes essential viral coat proteins. TRV is advantageous for its wide host range, high silencing efficiency, and long duration; thus, the TRV vector has been widely used in a variety of plants, including strawberry (*Fragaria × ananassa* Duch.) [23], cotton (*Gossypium* spp.) [16], and lychee (*Litchi chinensis* Sonn.) [15]. TRV has become the most widely used virus in the construction of VIGS systems.

VIGS technology is expected to overcome the bottlenecks in the field of tea tree research, such as the difficulty of achieving genetic transformation. However, some factors affect the infection efficiency. In general, a VIGS system can be divided into three steps. The first step is to construct the recombinant plasmid virus vector and build the silenced fragment of the target gene into the appropriate virus vector; the second step is to inoculate the recombinant virus vector into the recipient plant via various means; the last step is to evaluate the effect of the gene silencing by detecting various indicators. In previous studies, it was found that the length and location of the silencing fragment, the selection of the viral vector, the inoculation method of the recombinant viral vector, the concentration of the infection solution, the external environment of plant growth, different plant varieties, different growth periods, the duration of silencing, and the time of silencing all affect the infection efficiency of a VIGS system [11,22,24,25]. 

In this study, *CsPDS* was used as a marker gene to indicate the VIGS silencing effect [26]. Three injection sites and two cultivars were tested, and the results were assessed via chlorophyll content and the relative expression of *CsPDS*. We anticipate that TRV-based VIGS will be an effective tool for gene function analysis and for mining key genes for tea plant breeding.

## 2. Results

### 2.1. CsPDS Sequence Analysis

*CsPDS* was amplified from ‘LTDC’ according to the reference sequence (Figure 1). The full-length coding sequence (CDS) of *CsPDS* is 1749 bp, encoding 582 amino acids. Sequence alignment of *CsPDS* to PDS in other plants revealed several single-nucleotide polymorphisms (SNPs) that resulted in changes in amino acids among the different species (Figure 1). Analysis of the conserved domain revealed a special locus that belongs to the NAD_binding_8 superfamily, a large family of proteins that share a Rossmann-fold NAD(P)H/NAD(P)(+) binding (NADB) domain. The NADB domain is found in numerous dehydrogenases in metabolic pathways, such as glycolysis, and many other redox enzymes. NAD binding involves hydrogen bonds and van der Waals contacts, in particular the hydrogen bonding of residues in a turn between the first strand and the subsequent helix of the Rossmann fold topology. Characteristically, this turn exhibits a similar consensus binding pattern to GXGXXG, in which the first two glycines participate in NAD(P)-binding, and the third facilitates close packing of the helix to the beta strand. Typically, proteins in this family contain a second domain in addition to the NADB domain that is responsible for the specific binding to a substrate and catalysis of a particular enzymatic reaction.

### 2.2. pTRV2-CsPDS Vector Construction

Tobacco rattle virus (TRV) is a positive-strand RNA virus with a bipartite genome that is widely used for VIGS [27]. RNA-1 contains genes that are required for viral replication and transport, while RNA-2 encodes viral coat proteins [26]. TRV RNA-1 and RNA-2 cDNA were cloned into the *A. tumefaciens* binary vector system to generate pTRV1 and pTRV2 for plant transformation. The optimal siRNA, which should be 150–400 base pairs long with a GC content from 30 to 70%, was used [27]. The best-silencing site was selected using the online prediction tool pssRNAit, which predicted 9 siRNAs and 144 off-targets for *CsPDS*. The designed silencing gene fragment (SGF) for *CsPDS*, which was 200 bp in length with a GC content of 41%, was used to construct the pTRV2-*CsPDS* silencing plasmid (Figure 2a,b). The recombinant plasmid pTRV2-*CsPDS* was verified via gel electrophoresis and sequencing. As expected, the DNA fragment was confirmed to be ligated into the pTRV2 vector, based on the 200 bp shift relative to the empty vector control (Figure 2c).

### 2.3. Silencing of CsPDS in the Tea Plant

To determine the most effective injection site for infiltration, different sites were tested, including on the leaf back, petiole, and stem, and we found that the infection solution was able to infiltrate into the entire tea leaf using petiole injection (Figure 3a, middle). Leaf back injection can lead to necrotic lesions that cause the leaves to drop, which was not observed with petiole injection. A total of 22 tea plants (11 each) from two tea cultivars ‘LTDC’ and ‘YSX’ were infected with the *A. tumefaciens* strain GV3101 harboring pTRV1 and pTRV2-*CsPDS*, named PDS1–PDS11, respectively. After infiltration, a chlorosis symptom was observed in pTRV2-*CsPDS*-infected leaves, while leaves infected with the empty pTRV2 vector and uninfected control plants showed normal phenotypes (Figure 3b).

While a few of the infected plants began to exhibit chlorosis symptoms 3 days after infection (DAI), the chlorosis phenotype emerged 7 to 14 DAI in the majority of the plants. The majority of the chlorosis in ‘LTDC’ leaves occurred 7 DAI, while the chlorosis period of ‘YSX’ was 14 DAI. The chlorophyll content was assessed preliminarily with a handheld chlorophyll meter, and the data revealed that a total of 15 plants showed chlorosis symptoms. The chlorophyll contents of the leaves of nine ‘LTDC’ plants (PDS1, PDS2, PDS3, PDS4, PDS5, PDS6, PDS7, PDS8, and PDS10) 7 DAI (Figure 3c) and six ‘YSX’ plants (PDS3, PDS4, PDS7, PDS8, PDS9, and PDS11) 14 DAI (Figure 3d) were significantly decreased. Therefore, the silencing efficiency was cultivar-dependent; while ‘LTDC’ had an 81.82% success rate, that of ‘YSX’ was only 54.55%, which implies that ‘LTDC’ has a higher VIGS efficiency than ‘YSX’ (Figure 3e).

PCR was used to detect TRV1 and TRV2, and the results showed that pTRV1 and pTRV2 vectors were found in the photobleached plants and the empty vector group, but no target bands were found in the control group (Figure 3f), which implies that the vectors had been successfully transformed into the tea plants.

### 2.4. Evaluation of the Silencing Effect on Photobleached Leaves

Quantitative PCR was used to measure *CsPDS* expression at the transcription level to investigate whether the chlorosis symptom was caused by silencing *CsPDS*. We discovered that VIGS via petiole injection effectively silenced *CsPDS* in tea leaves (Figure 4b,f), and the chlorophyll concentration correlated with *CsPDS* gene transcription (Figure 4b–e,g–j). Statistical analysis revealed that the relative expression of *CsPDS* in silenced plants that exhibited the chlorosis symptom was significantly lower than that in the empty pTRV2 vector and non-injection control groups: relative to the empty vector, the expression of the *CsPDS* gene in ‘LTDC’ and ‘YSX’ silenced plants was comparable, decreasing by 53.66–90.24% and 59.57–95.74%, respectively. These results indicate that VIGS efficiency in tea plants may be genotype-independent and individual-plant-dependent.

The knockdown of *CsPDS* was further verified via the quantification of the chlorophyll content. The chlorophyll content of the two tea cultivars significantly decreased after the VIGS of the *CsPDS* gene (Figure 4c–e,h–j). Compared with the empty vector control, the chlorophyll contents in silenced ‘LTDC’ and ‘YSX’ plants decreased by 14.11–61.07% and 59.36–72.85%, respectively, which was consistent with the qRT-PCR results. In the PDS4 of ‘LTDC’, the expression level of *CsPDS* was downregulated to 10% relative to the empty vector, accompanied by a 42% reduction in chlorophyll (Figure 4a–e). The chlorosis symptom was also obvious in the PDS8 of ‘YSX’, where *CsPDS* expression and chlorophyll content were 4% and 36% of those in the empty vector, respectively (Figure 4f–j). These results demonstrate that the pTRV1 and pTRV2 VIGS system efficiently executed gene silencing in these tea plants.

## 3. Materials and Methods

### 3.1. Plant Materials

Two-year-old seedlings of two tea plant cultivars, i.e., ‘Ling Tou Dan Cong’ (‘LTDC’) and ‘Ya Shi Xiang’ (‘YSX’), were planted in the tea plant germplasm resource nursery of South China Agricultural University (Guangzhou, China). When the tea plants grew to one bud and four leaves, they were transferred to a greenhouse (24 °C, relative humidity 60–70%, 12 h/12 h light/dark) for subsequent infection. Plant leaves with an obvious chlorosis phenotype were used as the studied materials.

### 3.2. CsPDS Cloning

Total RNA was extracted from the young leaves of ‘LTDC’, and the full-length coding sequence of *CsPDS* was amplified from ‘LTDC’ cDNA with primers designed according to the reference sequence obtained from the Tea Plant Information Archive (TPIA) database (http://tpia.teaplants.cn/) [28]. The full-length *CsPDS* was ligated into a pMD19-T vector (Takara Biomedical Technology, Beijing, China), and the resulting recombinant plasmid was sequenced by Beijing Qingke Biotechnology Co., Ltd. (Beijing, China).

### 3.3. Vector Construction

The full-length coding sequence of *CsPDS* was submitted to the bioinformatics tool ‘pssRNAit’ (https://www.zhaolab.org/pssRNAit/Analysis.gy) (accessed on 20 October 2020) to find the optimal region from which to design a silencing fragment [29]. The *CsPDS* fragment was ligated to the pTRV2 vector via *EcoR*I and *BamH*I restriction sites using the homologous recombination technique, according to the instructions of the ClonExpress^®^II One Step Cloning kit C112 (Vazym, Nanjing, China). The primers are listed in Table 1. The pTRV1 and pTRV2 vectors were kindly supplied by the pepper research group at South China Agricultural University [30].

### 3.4. Agroinfiltration of Tea Plant Seedlings

The recombinant plasmid was transformed into *Agrobacterium tumefaciens* GV3101. *A. tumefaciens* GV3101 was cultured in 25 mL LB medium (containing 50 mg/L kanamycin and 25 mg/L rifampin) at 28 °C with shaking at 200 rpm until OD_600nm_ = 1. *A. tumefaciens* cells were then collected via centrifugation and resuspended in infiltration buffer containing 10 mM 2-N-morpholino ethanesulfonic acid (MES) and 10 mM MgCl_2_, and 200 µM acetosyringone. Then, 25 mL pTRV1 (OD600nm = 1.0) was mixed with pTRV2-CsPDS (OD600nm = 1.0) and pTRV2 (OD600nm = 1.0) in a ratio of 1:1, resulting in two groups of 50 mL mixed bacterial solution (pTRV1+pTRV2-*CSPDS*; pTRV1+pTRV2). In one bud at the three- or four-leaf stage of the tea plant, 1 mL of the mixed infection solution was injected into the tea leaf through the petiole, leaf back, or stem using a syringe (petiole and stem injections were also performed using a syringe with a needle, and leaf back injections were performed without needle). The infective liquid penetrated into the tea leaves completely. After infiltration, the plants were cultivated at 24 °C in the dark for 24 h, then transferred to normal cultivation conditions (24 °C, relative humidity 60–70%, 12 h/12 h light/dark) until the leaves showed obvious phenotypic changes and were collected for subsequent experiment.

### 3.5. Identification of Infected Plants

Leaf cDNA was utilized as a template to amplify pTRV1 and pTRV2. The empty vector served as a control. The coat protein (CP) of pTRV1 and pTRV2 was detected in injection plants via primers pTRV1-CP and pTRV2-CP, respectively, and the primers used are listed in Table 1.

### 3.6. Quantification of Chlorophyll A/B

A SPAD-502 plus chlorophyll meter (Konica Minolta, Japan) was used to preliminarily evaluate the contents of photosynthetic pigments. Extracted chlorophyll a and b were also quantified, as previously described [31]. Briefly, the tea leaves were cut into pieces, and 0.2 g samples were placed in 5 mL o extraction solution (acetone: 95% ethanol, in a 1:2 ratio) for 5 h in the dark. The absorbance was measured at 645 nm and 663 nm with a UV-550PC spectrophotometer (M&A instruments, Arcadia), and the extract solution was used as a reference. The chlorophyll a and b contents were calculated as follows:C_a_ = ((12.00 × A_663_) − (2.69 × A_645_)) × V/1000 W
C_b_ = ((22.70 × A_645_) − (4.68 × A_663_)) × V/1000 W
C_a+b_ = C_a_ + C_b_
where C_a_ and C_b_ are chlorophyll a and chlorophyll b (mg/g); C_a+b_ is the total chlorophyll (mg/g) content; A_663_ and A_645_ refer to the absorbance of chlorophyll a and b at 663 nm and 645 nm, respectively; V is the volume of the extract; and W is the weight of fresh leaves.

### 3.7. Quantitative Real-Time Reverse Transcription PCR (qRT-PCR)

Infected tea plant leaves were harvested and immediately frozen in liquid nitrogen. RNA was extracted using a HiPure Plant RNA Mini Kit b (Magen, China). The RNA was reverse-transcribed into cDNA using a HiScriptRIII RT Super Mix for qPCR (+gDNAwiper) (Vazyme, China) according to the manufacturer’s instructions. qRT-PCR was carried out according to our previous method [32]. *Actin* was used as a reference gene, and the relative expression levels were calculated using the 2^−ΔΔCT^ method. The primers used are listed in Table 1.

### 3.8. Statistical Analysis

SPSS Statistics 17.0 and Excel (Microsoft Office 2019) were used to calculate statistics. The least significant difference (LSD) method was utilized to determine whether any samples had significant differences. All the data are presented as the mean of three replicates.

## 4. Discussion and Conclusions

Genes that disrupt the synthesis or the stability of chlorophyll, anthocyanin, or carotenoid pigments are often adopted as marker genes in VIGS systems for their readily visible phenotypes [26,33,34]. The PDS gene encodes an octa-hydro lycopene dehydrogenase that catalyzes carotenoid production, which, under normal circumstances, protects chlorophyll from photolysis. Therefore, silencing the PDS gene in plants results in photobleaching symptoms and chlorosis symptoms due to light-induced chlorophyll degradation. VIGS systems employing homologous PDS genes have been successfully established in plants such as tobacco, potato, cotton, and cannabis due to this clear phenotypic trait [10,22,35,36]. Through sequence analysis, we found that the CsPDS gene is highly conserved in tea plants, and silencing this gene with a single construct had the same effect in two tea cultivars. The enzyme protochlorophyllide oxidoreductase (POR) catalyzes a light-dependent step in chlorophyll biosynthesis that is essential to photosynthesis [37]. Recently, the TRV-VIGS system was used to silence CsPOR, and photobleaching symptoms were observed [38]. These studies indicate that the silencing of either CsPDS or CsPOR leads to photobleaching symptoms, and each could serve as a visible phenotype for developing and evaluating the VIGS system in tea plants.

Many VIGS vectors have been developed, including TRV, cucumber mosaic virus (CMV), barley stripe mosaic virus (BSMV), and rice tungro bacilliform virus (RTBV) [39,40]. The choice of a vector for VIGS is species-dependent and is chosen based on the virus infection ability and silencing efficiency. The TRV-derived VIGS vector has been widely used in different dicotyledon plants, such as those in *Solanaceae* [14], *Gossypium hirsutum* [41], *Litchi chinensis* [15], apple (*Malus pumila*) [42], and pear (*Pyrus* spp.) [43]. In this study, we selected the TRV-based VIGS vector, and high silencing efficiency was achieved. This indicated that the TRV vector was able to be applied in *C. sinensis* for the downregulation of genes by transiently introducing the viral vector containing a gene-silencing fragment into tea plant cells via agroinfiltration. However, it is difficult to infiltrate *A. tumefaciens* into woody plant leaves. Previous studies indicated that infiltration via injection into the back of the leaves is suitable for VIGS in most plants [14]; this method can elicit a strong immunoreaction and cause the leaves to drop, and the same results were observed in our study. A recently published study was successful at silencing the gene expression in the tea plant via the vacuum infiltration method [37]. In this study, we optimized the injection method and found that petiole injection is the easiest and most effective method of delivering *A. tumefaciens* to the tea leaves (Figure 3a).

The *Agrobacterium* solution plays an important role in the efficiency of VIGS. In this study, we found that using *A. tumefaciens* at an OD600 of 1.0 led to considerable silencing efficiency. Other studies have found that an OD600 of 1.2 was best for the infiltration of the tea plant [38], which implies that a high concentration of *A. tumefaciens* might be crucial for silencing efficiency. Previous studies reported that, in infected plants cultivated at a room temperature ranging from 22 °C to 25 °C, the duration of gene silencing was increased [24]. In this study, the temperature was maintained at 24 °C, and we obtained high silencing efficiency, which is consistent with the data described in a previous study [39].

Three DAI, slight leaf chlorosis symptoms were observed, and a more apparent phenotype was observed from 7 to 14 DAI. Recently, the vacuum infiltration method was used to silence the chlorophyll biosynthesis gene *CsOPR1*, and photobleaching symptoms were observed 25 DAI [38]. Presumably, different genotypes, infection methods, and marker genes might lead to different results. We believe that higher silencing efficiency can be achieved when there is sufficient observation time [44]. Research shows that mild viral symptoms often appear 7–10 DAI, and that spreading to fresh, young plant leaves may take up to 2–3 weeks, with the most obvious phenotype being produced in 3 weeks [26].

In this study, a preliminary VIGS system was established for the tea plant. However, to further optimize the system and to find the ideal experimental settings, more in-depth investigation should be carried out to consider the selection of a suitable genotype, length, and GC content for the silencing segment, osmotic concentration of the infiltration solution, and silencing duration [44,45]. For example, we found that ‘YSX’ required a longer duration to achieve chlorosis symptoms than ‘LTDC’. Usually, the gene silencing phenotype induced by TRV lasts 1–2 months before gradually decreasing or disappearing [46], but the factors that affect the phenotype and silencing duration need further study. Nonetheless, the present study achieved the silencing of *CsPDS* in two tea cultivars and established that petiolar injection is better than back-of-leaf or stem injection for the infiltration of the entire leaf. These results provide a reference for the optimization of the tea VIGS system. A successful VIGS system will be a great asset to using reverse genetics in tea.

## Figures and Tables

**Figure 1 plants-12-03162-f001:**
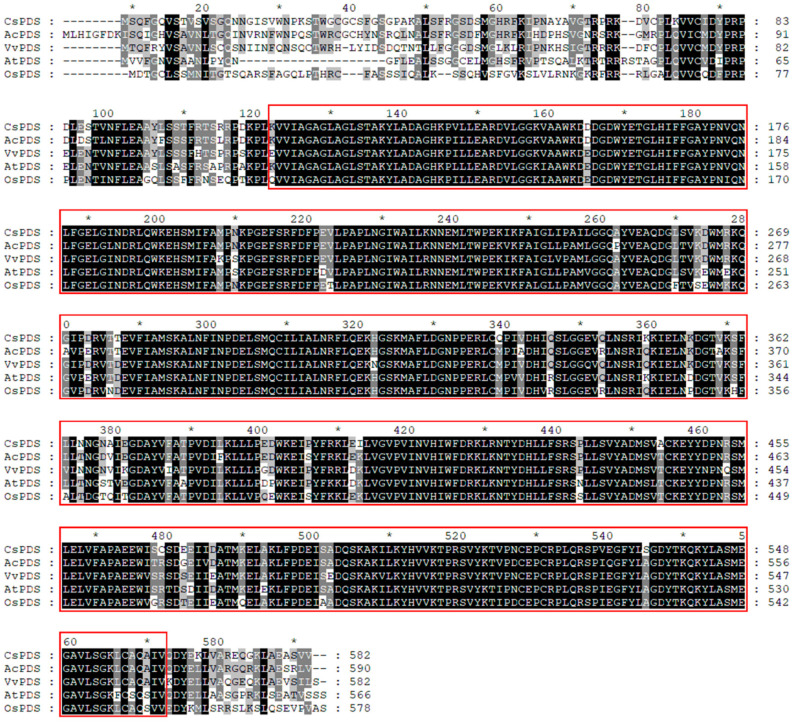
Conserved domains of *CsPDS*. The red boxes indicate the conserved domains of the PDS superfamily, and the * indicated singular ten number of amino acid sequence. The *PDS* sequences of *Actinidia chinensis var. chinensis*, *Arabidopsis thaliana*, *Oryza sativa* L., and *Vitis vinifera* L. were downloaded from NCBI (https://www.ncbi.nlm.nih.gov).

**Figure 2 plants-12-03162-f002:**
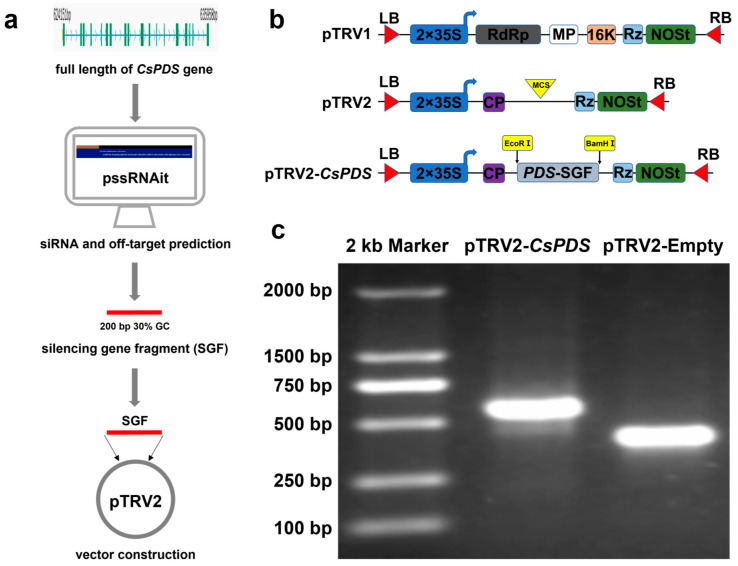
Schematic of pTRV2-*CsPDS* vector construction. (**a**) Map of pTRV1, pTRV2, and pTRV2-*CsPDS* vectors. LB, left border; RB, right border; 2 × 35S, two consecutive 35S promoters; RdRp, RNA-dependent RNA polymerase; MP, movement protein; 16K, 16 kD protein; Rz, self-cleaving ribozyme; NOSt, NOS terminator; CP, coat protein; and MCS, multiple cloning site. (**b**) Schematic of the construction process. (**c**) PCR amplification of the *CsPDS* silencing gene fragment (SGF) in the empty and recombinant plasmid.

**Figure 3 plants-12-03162-f003:**
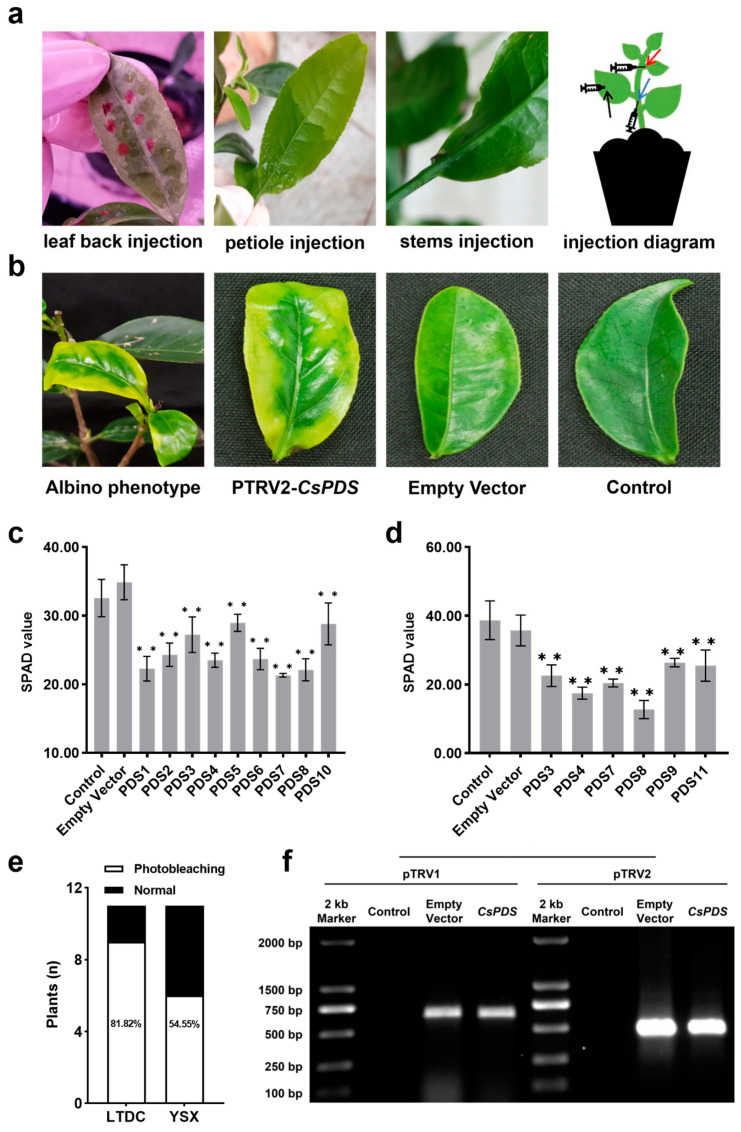
Silencing of *phytoene desaturase* (*PDS*) in *Camellia sinensis* L. (**a**) Injection sites indicated. Red arrow for petiole injection; blue arrow stem for injection; black arrow for leaf back injection; (**b**) chlorosis symptom in ‘YSX’ leaves after treatment with pTRV1+pTRV2-*CsPDS* and the controls; nondestructive evaluation of chlorophyll content using a SPAD-502 plus chlorophyll meter on infected leaves of ‘LTDC’ on the seventh day after injection (DAI) (**c**) and ‘TSX’ 14 DAI (**d**). Statistical analyses to compare the plants were carried out with the least significant difference (LSD) test (** *p* < 0.01). Data are presented as the mean ± SD (*n* = 3); silencing efficiency in ‘LTDC’ and ‘YSX’ (**e**); (**f**) detection of tobacco rattle virus (TRV) fragments after virus-induced gene silencing (VIGS) treatments.

**Figure 4 plants-12-03162-f004:**
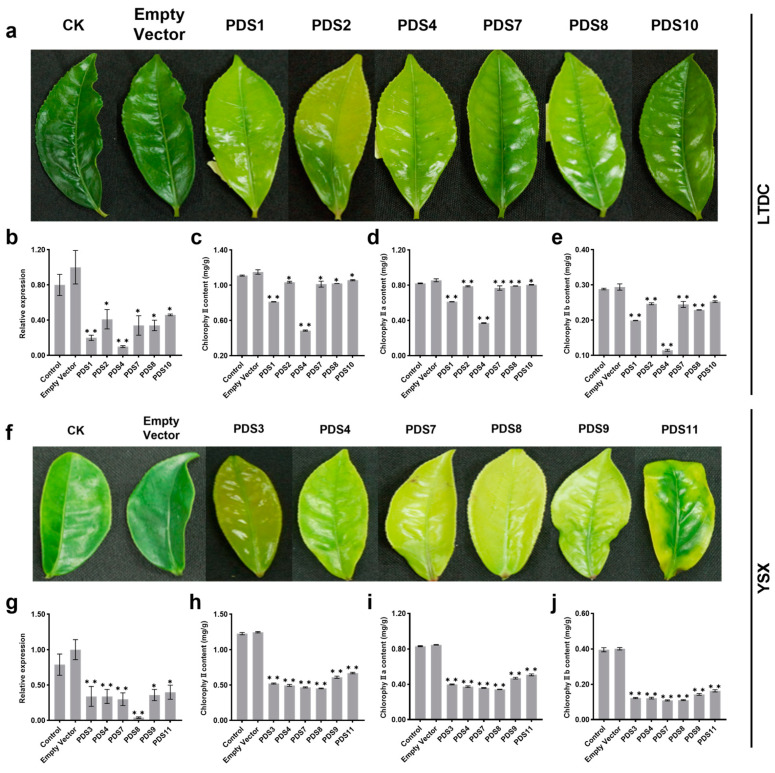
Evaluation of virus-induced gene silencing (VIGS) of *phytoene desaturase* (*PDS*) using petiole injection in tea leaves. Chlorosis symptoms after VIGS of *CsPDS* in ‘LTDC’ on the seventh day after injection (DAI) (**a**) and ‘YSX’ on the fourteenth DAI (**f**); each of 11 tea plants from ‘LTDC’ and ‘YSX’ were infected, named PDS1-PDS11, respectively, and (**a**,**e**) only presented the leaves of a single plant with chlorosis symptoms. Relative expression of *CsPDS* in ‘LTDC’ (**b**) and ‘YSX’ (**g**); total chlorophyll, chlorophyll a, and chlorophyll b contents in the VIGS-treated leaves of ‘LTDC’ (**c**–**e**) and ‘YSX’ (**h**–**j**). Statistical analyses to compare the plants were carried out with the least significant difference (LSD) test (* *p* < 0.05 and ** *p* < 0.01). Data are presented as the mean ± SD (*n* = 3).

**Table 1 plants-12-03162-t001:** List of primers used in this study.

Description	Primer Name	Primer Sequence
Gene cloning	CsPDS	F: ATGTCTCAATTTGGACAAGTTTCC
		R: TTACACGACACTTGCCTCGGCCA
Vector construction	pTRV2-CsPDS	F: agaaggcctccatggggatccGGAGTTCCTGTTATAAATGTTCACATATG
		R: tgtcttcgggacatgcccgggTCATCACTACATGAGATCCATTCCTC
Positive clones’ detection	pTRV2	F: TGAGGGAAAAGTAGAGAACG
		R: CCTATGGTAAGACAATGAGT
TRV virus verification	pTRV1-CP	F: TTACAGGTTATTTGGGCTAG
		R: CCGGGTTCAATTCCTTATC
TRV virus verification	pTRV2-CP	F: ATATTCCTGCGAATCCAAACAC
		R: GAA ACTCAAATGCTACCAACGA
qRT-PCR analysis	qCsPDS	F: TACTTCCCGCCGTCCAGATAAAC
		R: AAGCCAGTCTCATACCAGTCTCCAT
qRT-PCR analysis	qCsActin	F: GCCATATTTGATTGGAATGG
		R: GGTGCCACAACCTTGATCTT

## Data Availability

Not applicable.

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
