# Peer review of "Virus-Induced Gene Silencing in the Tea Plant (Camellia sinensis)"

_plants, 2023, doi:10.3390/plants12173162_

Round 1

Reviewer 1 Report

The ms. by Wei et al. reveals that VIGS is also a useful technique in molecular and physiological studies involving tea (Camellia sinensis) plants. The study is well-described and brings a detailed description of the methodology. The figures are well organized and support the obtained results. I would like to know whether the author tested the susceptibility of tea plants to TRV and how those results can be correlated with the VIGS obtained results in the two tea genotypes.

Some other minor issues to be amended:

Lines 3, 12: Please, scientific name in italics. Here and elsewhere in the text.

Line 57: Add scientific names of the described plants.

Lines 182-185: Mentioned colors in the legend of Figure 1 are missing. Similarly, the mentioned row compressing the consensus sequence is not shown. Verify!! Add the corresponding acronym for plant PDS genes, i.e.  AcPDS.

Line 241-242: “These results indicated that pTRV2-CsPDS exerted its function to silence the target gene.” This is an indirect inference. In my opinion, this sentence is conclusive and involves results shown in experiments presented later in the ms. Please, rewrite and avoid oversizing the result.

Line 244: In Figure 3f, bands derived from pTRV2 control and CsPDS seem to be of the same size. Could you verify this?

Author Response

12-July-2023

Dear reviewer:

Thank you for reviewing our manuscript on Plants. We appreciate the constructive comments and professional advice provided by you. These opinions have helped improve the academic rigor of our article. Below are your comments, followed by our point-by-point responses in blue.

Sincerely,

Binmei Sun

South China Agricultural University

Point-by-point responses to editor’s and reviewers’ comments

Reviewer 1

The ms. by Wei et al. reveals that VIGS is also a useful technique in molecular and physiological studies involving tea (Camellia sinensis) plants. The study is well-described and brings a detailed description of the methodology. The figures are well organized and support the obtained results.

Q1: I would like to know whether the author tested the susceptibility of tea plants to TRV and how those results can be correlated with the VIGS obtained results in the two tea genotypes.

R1: Thank you for your comments. In fact, A total of 22 tea plants (11 each) from two tea cultivars ‘LTDC’ and ‘YSX’ were infected with A. tumefaciens strain GV3101 harboring pTRV1 and pTRV2-CsPDS. Only 9 single plants of‘LTDC’ and 7 single plants of‘YSX’ exhibited a photobleaching phenotype, while the empty vector control and some tea plants didn’t show photobleaching phenotype. In the same time, the coat protein (CP) of Tobacco rattle virus was detectable in photobleaching phenotype plant, coincided with lower transcription of CsPDS and reduced chlorophyll content compared with the empty vector control. Above all, the photobleaching phenotype could be correlated with the VIGS effect.

Some other minor issues to be amended:

Q2: Lines 3, 12: Please, scientific name in italics. Here and elsewhere in the text.

R2: Thank you for your comments. The scientific names have been changed to italics in all manuscript.

Q3: Line 57: Add scientific names of the described plants.

R3: Thank you for your comments. The scientific names of described plants have been added (Line 62-64).

Q4: Lines 182-185: Mentioned colors in the legend of Figure 1 are missing. Similarly, the mentioned row compressing the consensus sequence is not shown. Verify!! Add the corresponding acronym for plant PDS genes, i.e.  AcPDS.

R4: Sorry for the mistake. We have deleted the colors description and modified the description in the legend of Figure 1 (Line 219-222).

Q5: Line 241-242: “These results indicated that pTRV2-CsPDS exerted its function to silence the target gene.” This is an indirect inference. In my opinion, this sentence is conclusive and involves results shown in experiments presented later in the ms. Please, rewrite and avoid oversizing the result.

R5: Thank you for your comments. We have modified our description (Line 277-284).

Q6: Line 244: In Figure 3f, bands derived from pTRV2 control and CsPDS seem to be of the same size. Could you verify this?

R6: Thank you for your comments. We have been double checked the results and it was correct. The bands derived from pTRV2 control and CsPDS were the same is because we detected the coat protein (CP) of Tobacco rattle virus instead of multiple cloning site (MCS). The CP will allow virion formation and transmission across plants, which were widely used for detecting the transformed vector. (Ref: 1. Liu et.al. 2015. The vectors have been successfully transformed into the tea plants. Scientia Agricultura Sinica, doi: 10.3864/j.issn.0578-1752. 2. Gongyao et.al. 2021. A Methodological Advance of Tobacco Rattle Virus-Induced Gene Silencing for Functional Genomics in Plants. Front Plant Sci. 2021; 12: 671091.)

Reviewer 2 Report

Dear authors,

please check:

Line 280 – 288: check the caption of the figure 4. It is not clear; in my opinion (e) should be (f), etc. 

And it is not clear what are plants PDS1 – PDS 11 (should be explained). If these are biological repetitions, please indicate

Please check Figure 3d, where the empty vector (negative control) has a large range of variability and PDS 2 and 9 (silenced) have values within it, but the differences are shown as statistically significant. Is this correct?

The same issue appears in Figure 4d, where silenced plants don't seem significantly different from controls.

Or explain these issues in discussion or involve the expert for statistics and re-analyze the data.

Author Response

12-July-2023

Dear reviewer:

Thank you for reviewing our manuscript on Plants. We appreciate the constructive comments and professional advice provided by you. These opinions have helped improve the academic rigor of our article. Below are your comments, followed by our point-by-point responses in blue.

Sincerely,

Binmei Sun

South China Agricultural University

Point-by-point responses to editor’s and reviewers’ comments

Reviewer 2

Dear authors,

please check:

Q1: Line 280 – 288: check the caption of the figure 4. It is not clear; in my opinion (e) should be (f), etc.

And it is not clear what are plants PDS1 – PDS 11 (should be explained). If these are biological repetitions, please indicate

R1: Sorry for the mistake. (e) have been changed to (f) and PDS1 – PDS 11 have been explained in the caption of the figure 4 (Line 327-329).

Q2: Please check Figure 3d, where the empty vector (negative control) has a large range of variability and PDS 2 and 9 (silenced) have values within it, but the differences are shown as statistically significant. Is this correct?

The same issue appears in Figure 4d, where silenced plants don't seem significantly different from controls.

Or explain these issues in discussion or involve the expert for statistics and re-analyze the data.

R2: Thank you for your comments. We have been re-analyzed the data and double checked the results. The Figure 3d and Figure 3e has been modified. In figure 4d, the significant difference was compared to empty vectors instead of controls.
